RESEARCH COMMUNICATION

# BRCT domains of the DNA damage checkpoint proteins TOPBP1/Rad4 display distinct specificities for phosphopeptide ligands

**Matthew Day[1], Mathieu Rappas[1†], Katie Ptasinska[2], Dominik Boos[3], Antony W Oliver[1]\*, Laurence H Pearl[1]\***

[1]Cancer Research UK DNA Repair Enzymes Group, Genome Damage and Stability Centre, School of Life Sciences, University of Sussex, Falmer, United Kingdom; [2]Genome Damage and Stability Centre, School of Life Sciences, University of Sussex, Falmer, United Kingdom; [3]Fakultät für Biologie, Universität Duisburg-Essen, Germany, United Kingdom

**\*For correspondence:**
antony.oliver@sussex.ac.uk (AWO);
laurence.pearl@sussex.ac.uk (LHP)

**Present address:** [†]Heptares Therapeutics Ltd., Cambridge, United Kingdom

**Competing interests:** The authors declare that no competing interests exist.

**Abstract** TOPBP1 and its fission yeast homologue Rad4, are critical players in a range of DNA replication, repair and damage signalling processes. They are composed of multiple BRCT domains, some of which bind phosphorylated motifs in other proteins. They thus act as multi-point adaptors bringing proteins together into functional combinations, dependent on post-translational modifications downstream of cell cycle and DNA damage signals. We have now structurally and/or biochemically characterised a sufficient number of high-affinity complexes for the conserved N-terminal region of TOPBP1 and Rad4 with diverse phospho-ligands, including human RAD9 and Treslin, and *Schizosaccharomyces pombe* Crb2 and Sld3, to define the determinants of BRCT domain specificity. We use this to identify and characterise previously unknown phosphorylation-dependent TOPBP1/Rad4-binding motifs in human RHNO1 and the fission yeast homologue of MDC1, Mdb1. These results provide important insights into how multiple BRCT domains within TOPBP1/Rad4 achieve selective and combinatorial binding of their multiple partner proteins.
**Editorial note:** This article has been through an editorial process in which the authors decide how to respond to the issues raised during peer review. The Reviewing Editor's assessment is that all the issues have been addressed (see decision letter).
DOI: https://doi.org/10.7554/eLife.39979.001

## Introduction

TOPBP1 and its fission and budding yeast orthologues Rad4 and Dpb11, respectively, are scaffold proteins that mediate formation of multi-protein complexes in a range of essential DNA replication and repair processes (*Wardlaw et al., 2014*). Homologues in vertebrates each contain nine BRCT domains, of which four (at the N-terminus; BRCT1,2 and 4,5) are conserved in the yeasts (*Garcia et al., 2005*). In addition, all members of the family possess a domain required for activation of the DNA damage PI3-kinase-like kinase ATR (*Kumagai et al., 2006*; *Lin et al., 2012*; *Mordes et al., 2008a*; *Mordes et al., 2008b2008b*). In TOPBP1, this is located between BRCT domains 6 and 7, while in the yeast proteins it occurs at the C-terminus, downstream of BRCT4 (*Wardlaw et al., 2014*).

Bioinformatic analysis (*Rappas et al., 2011*; *Wardlaw et al., 2014*) suggests that only three of the BRCT domains in the yeast TOPBP1 homologues, and four in the vertebrate TOPBP1 homologues, possess the required cluster of residues needed to bind phosphorylated peptide motifs.

Despite the structural similarity of these phosphopeptide-binding BRCT domains, genetic and biochemical studies show that they are nonetheless selective and specific for the phosphopeptide sequences with which they interact (*Leung et al., 2011*; *Leung et al., 2013*; *Qu et al., 2013*; *Rappas et al., 2011*; *Sun et al., 2017*). However, the structural basis for this specificity has not been defined.

To gain some insight into this, we have determined crystal structures of segments of yeast and vertebrate TOPBP1 that contain the two most N-terminal phosphopeptide binding sites (BRCT1 and BRCT2), in complex with phosphopeptides derived from a number of different known ligand proteins, including RAD9, and used this to develop consensus patterns that encapsulate the individual specificity requirements of these two independent binding sites. We have used these patterns to predict interacting phospho-peptides from putative ligand proteins in vertebrates and yeast, and have characterised their specific interactions structurally and biochemically. These data provide insights into how TOPBP1 homologues can differentiate between closely related phosphopeptide motifs, allowing them to act as multi-point scaffolds that bring ligand proteins together into specific functional assemblies for DNA replication and repair.

## Results

### Structural basis for RAD9 binding to TOPBP1

Phosphorylation of RAD9 on Ser387 (or its equivalent) in its C-terminal 'tail' has been shown to be essential for interaction with TOPBP1, and required for an effective checkpoint response in a *Xenopus* system (*Lee et al., 2007*). Subsequently, this phosphorylation was shown to be dependent on casein kinase 2 (CK2) (*Rappas et al., 2011*; *Takeishi et al., 2010*) and its site of interaction on TOPBP1 localised to BRCT1, with an affinity in the low micromolar range ($K_d$ = 2.1 μM) (*Rappas et al., 2011*). However, the structural basis for the interaction of RAD9-pS387 with TOPBP1, and how this directs specificity to just one out of the nine BRCT domains in TOPBP1 was unclear.

Although we previously determined the crystal structure of the N-terminal segment (BRCT0,1,2) of human TOPBP1 (*Rappas et al., 2011*), no structures have been reported for peptide complexes of this region. We have now succeeded in determining the crystal structure of the BRCT0,1,2 module of chicken TOPBP1 (cTOPBP1) in complex with a peptide corresponding to the phosphorylated C-terminus of human RAD9 at a resolution of 2.3 Å.

As determined by mutagenesis studies (*Rappas et al., 2011*), the RAD9-pS387 peptide binds to the middle BRCT domain (BRCT1) of the closely packed array that forms the N-terminus of cTOPBP1 (*Figure 1A*). Unlike phospho-peptide binding to the more commonly occurring C-terminal tandem BRCT$_2$ arrangements (*Baldock et al., 2015*; *Kilkenny et al., 2008*; *Leung et al., 2011*; *Rodriguez et al., 2003*; *Stucki et al., 2005*) where the bound peptide bridges consecutive BRCT domains, the RAD9-pS387 peptide runs perpendicular to the long axis of the BRCT-domain 'array' and only contacts BRCT1 (*Figure 1B*).

The phosphate group of RAD9-pSer387 is recognised by a network of hydrogen bonding and ionic interactions centred around the side chains of cTOPBP1 residues Thr114, Arg121 and Lys155, which form the conserved triplet common to all BRCT-domains involved in binding phosphorylated peptide motifs (*Wardlaw et al., 2014*). In addition to these core polar interactions with the phosphate, the side chain of RAD9-Asp386 hydrogen bonds to the side chain of cTOPBP1-Lys118. No ordered electron density was observed for the downstream Glu-Gly-Glu-Gly sequence that forms the C-terminal tetrapeptide of RAD9. Upstream of pSer387 and Asp386 (−1, relative to phosphorylated residue), the peptide backbone of the RAD9 peptide main chain forms a distorted anti-parallel β-sheet with the main chain of a loop connecting cTOPBP1 residues 137 – 141, with an additional interaction between the side chain of cTOPBP1-Asp138 and the peptide nitrogen of RAD9-Asp386 (*Figure 1C*). The effect of these main chain interactions is to twist the peptide into a tight turn, which projects the sequential side chains of RAD9 residues Leu383 (−4) and Ala384 (−3) into a hydrophobic pocket lined by the side chains of cTOPBP1 residues Lys154, Lys155, Val158, Leu162, Leu139 and Met141. All the cTOPBP1 residues involved in binding the RAD9 peptide are identical in the human protein, except Met141 which is a valine in human TOPBP1. The tight turn conformation of the RAD9 peptide is reinforced by the hydrophobic side chain of Val382 which packs in against the face of the peptide bond connecting residues Leu383 and Ala384.

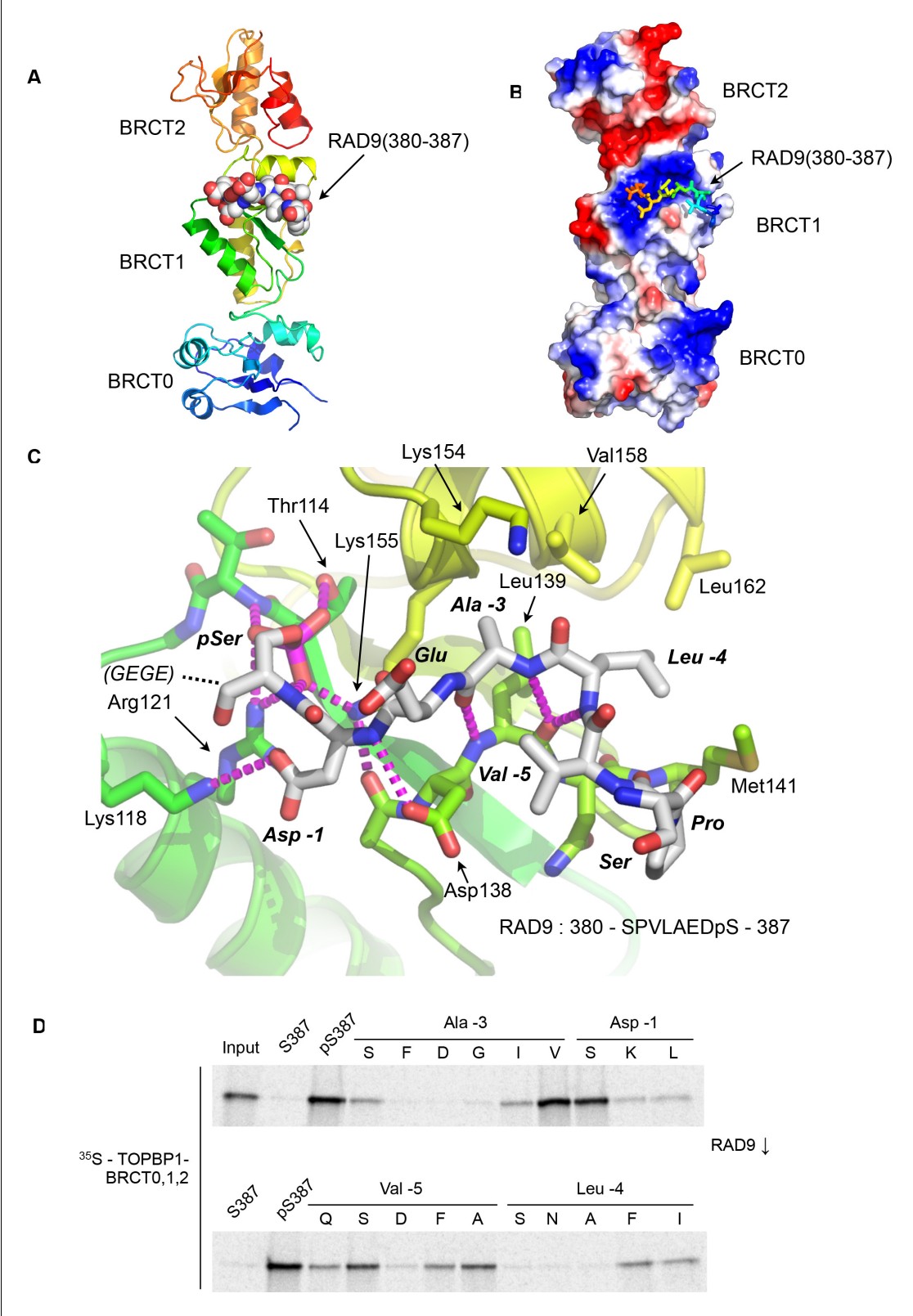

**Figure 1.** Crystal structure of the TOPBP1 – RAD9 interaction. (A) Secondary structure cartoon (rainbow coloured N:blue → C:red) of the BRCT0,1,2 module of chicken TOPBP1 bound to the C-terminal tail of human RAD9 (CPK model) phosphorylated on Ser387. (B) The RAD9-pS387 peptide binds to a positively charged patch (blue) on BRCT1 of TOPBP1 (C) Close-up of the TOPBP1 - RAD9 interaction. The negatively charged phosphate group of RAD9-pSer387 makes multiple hydrogen binding interactions with residues that are strongly topologically conserved in all phosphopeptide-binding

*Figure 1 continued on next page*

*Figure 1 continued*

BRCT domains. An additional polar interaction is provided by RAD9-Asp386. The consecutive hydrophobic side chains of RAD9-Leu383 and Ala384 insert into a pocket in TOPBP1-BRCT1, enabled by their main chains packing against the side chain of Val382, in a tight turn conformation. (D) Sub-site preferences of RAD9 tail binding to TOPBP1-BRCT1. Biotinylated peptides based on the C-terminal residues of *Hs*RAD9 with or without phosphorylation on Ser387, and pS387 peptides with single point mutations in positions −1, −3, −4 and −5 relative to the phosphorylated residue, were used to pull-down radiolabelled in vitro translated TOPBP1-BRCT0,1,2 (see Materials and methods), with the relative yields of bound protein in the autoradiographs reflecting which amino acids can be accommodated at the different positions in the bound peptide sequence (see text).

DOI: https://doi.org/10.7554/eLife.39979.002

To determine which RAD9 residues, in addition to the phosphorylated Ser 387, are important for binding to TOPBP1, we established a semi-quantitative peptide pull-down assay in which we systematically varied the identity of the residues at position −5, −4, −3, and −1 relative to pSer387 (*Figure 1D*). Position −3 (Ala in RAD9) retained tight binding only when substituted by valine, while −4 (Leu in RAD9) tolerated phenylalanine or isoleucine, but with reduced binding in all cases. Position −5 (Val in RAD9), showed the greatest tolerance, retaining substantial binding with a range of substitutions, including proline and polar residues such as serine or glutamine. The side chain at this position is conformationally unrestricted beyond the Cγ position and involved primarily in interactions within the RAD9 peptide itself that reinforce the tight turn conformation delivering the side-chains of residues −3 and −4 into direct interaction with TOPBP1. The unfavourability of aspartic acid at this position is likely to result from a repulsive electrostatic interaction with TOPBP1-Asp138. The aspartic acid in the −1 position, which makes a polar interaction with a lysine residue in TOPBP1, could be replaced by serine without substantial loss of binding, but replacement with leucine or lysine, which would generate a steric and/or electrostatic clash, abolished binding.

## Conserved conformations of TOPBP1/Rad4 binding peptides

Although no crystal structures have been reported for phospho-peptides binding to BRCT2 of metazoan TOPBP1, we previously described the interaction of phospho-peptides from Crb2 (the fission yeast homologue of 53BP1) to both BRCT1 and BRCT2 of Rad4 (the fission yeast homologue of TOPBP1) (*Qu et al., 2013*). We have now also determined the crystal structure of a bis-phosphorylated pT636-pT650 peptide from Sld3 (the fission yeast homologue of Treslin), which interacts simultaneously with Rad4-BRCT1 and Rad4-BRCT2, albeit with different molecules in the asymmetric unit of the crystals (*Figure 2A*).

Structural alignment of the Rad4 – Sld3-pT636,pT650 complex with the previously described Rad4 – Crb2-pT187 complex, and the newly described TOPBP1 – RAD9-pS387 complex, reveals the presence of the same tight turn conformation adopted by the RAD9 peptide (see above) in all the peptides bound to BRCT1 (*Figure 2B*). Their respective amino acid sequences reflect this propensity by the strong conservation of a hydrophobic triplet ending three residues upstream of the phospho-threonine.

Conversely, structural alignment of previously described complexes of Rad4 (*Qu et al., 2013*) in which peptides are instead bound to BRCT2 (Rad4 - Crb2-pT187 and Rad4 - Crb2-pT235) with the Rad4 – Sld3-pT636,pT650 complex, confirms a different conformation for the ligand (*Figure 2C*). In these complexes, the peptide backbone upstream of the phosphorylated residue maintains a consistent β-sheet conformation, and does not contain the tight turn seen for the hydrophobic triplets in peptides bound to the BRCT1 domain of TOPBP1 or Rad4.

## Structural basis for peptide selectivity by BRCT1 and BRCT2

Comparison of the BRCT1 and BRCT2 domains of TOPBP1 and of Rad4 suggests conserved features that govern the different conformation of their ligand peptides and contribute to selectivity. A common feature of all TOPBP1/Rad4 BRCT1 or BRCT2 ligands so far identified, is the presence of a hydrophobic residue at −3 relative to the phosphorylated serine or threonine. In both BRCT1 and BRCT2, this residue packs into a hydrophobic recess formed by side chains projecting from the α-helix connecting the 3rd and 4th β-strand of the BRCT fold. In BRCT1 this recess is extended and is sufficiently large to accommodate two consecutive hydrophobic side chains of a peptide ligand so long as the backbone of the peptide adopts a tight turn conformation (*Figure 3A,B*). In BRCT2, however, the size of the recess is restricted by the side chain of a highly conserved tryptophan

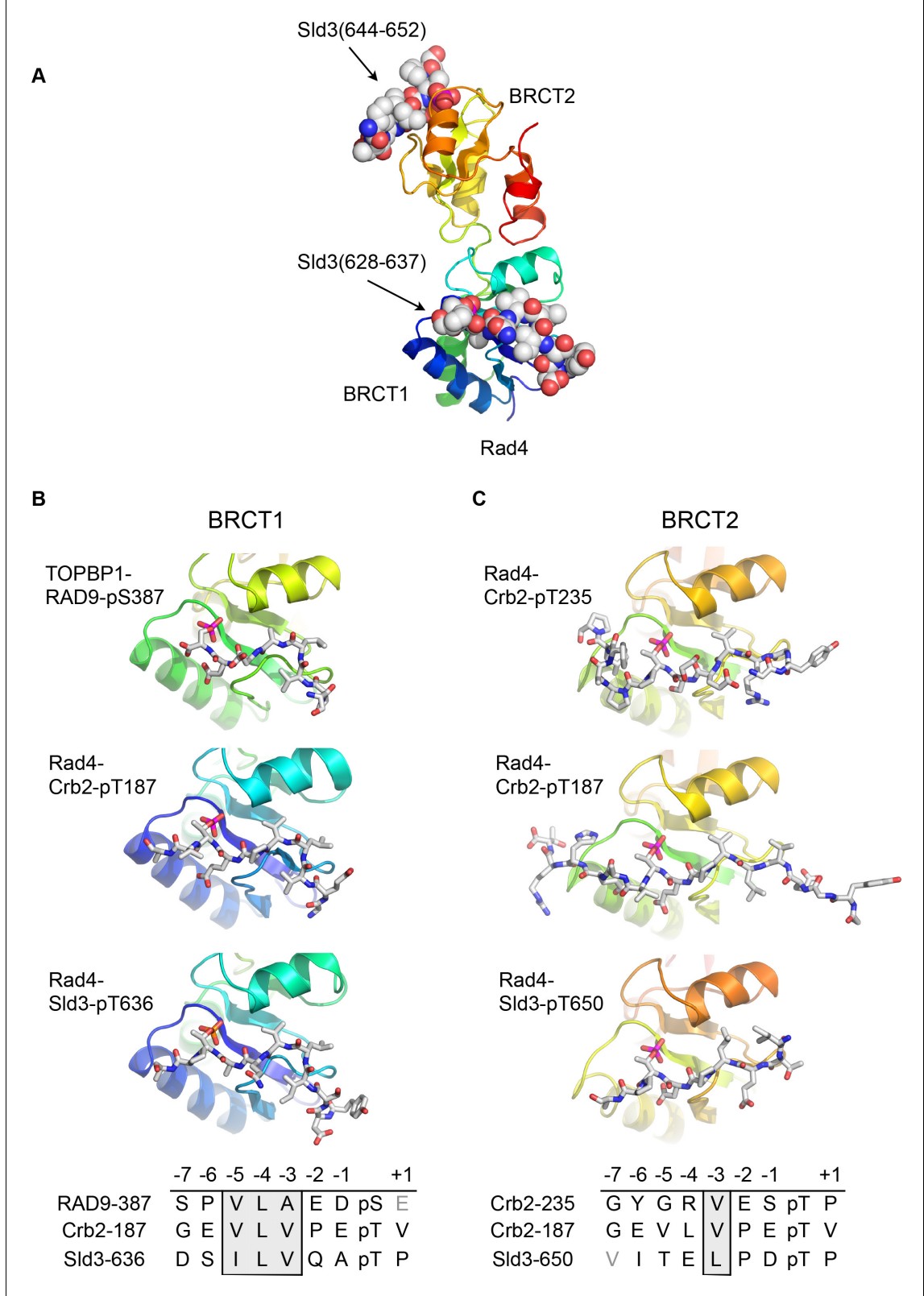

**Figure 2.** Conserved conformations of BRCT1 and BRCT2 ligands. (**A**) Crystal structure of the BRCT1,2 module of Rad4, the fission yeast homologue of TOPBP1, bound simultaneously to peptides from Sld3, the fission yeast homologue of Treslin, phosphorylated on Thr636 and on Thr650. (**B**) Montage of crystal structures of phosphopeptide complexes with BRCT1 of TOPBP1 or Rad4 – the sequences of the peptides is shown below. The other BRCT domains present in the crystals are omitted for clarity. All three examples share a common tight turn conformation for the −5, −4 and −3 region, as

*Figure 2 continued on next page*

*Figure 2 continued*
described above for TOPBP1-RAD9. (C) As B, but for BRCT2. Unlike BRCT1, the −5, −4, −3 regions of the bound peptides have an extended backbone conformation, in which only the hydrophobic side chain at position −3 binds into a pocket on the BRCT domain.
DOI: https://doi.org/10.7554/eLife.39979.003

residue (Trp158 in Rad4; Trp257 in human TOPBP1) at the C-terminal end of the α-helix, and can only accommodate a single hydrophobic side chain, requiring that the peptide chain continues in an extended β-sheet conformation (*Figure 3C,D*).

Based on the available crystal structures, the major factor that determines binding to BRCT1 seems to be the ability of the peptide sequence upstream of the phosphorylated residue to form a tight turn that places the side chains of the −3 and −4 residues into the extended hydrophobic recess. Sequences with hydrophobic residues at −3 and −4 (and often −5) as in the RAD9-pS387, Crb2-pT187, Sld3-pT636 and Treslin-pS1001 (993-DIGVVEEpSP) peptides conform to this requirement, whereas sequences with large polar side chains such as Crb2-pT235 where −4 is arginine, or Sld3-pT650 where −4 is glutamate, do not, and are restricted to binding to BRCT2. A subtler effect is seen with the Treslin-pT969 site (962-LTKSVAEpTP), which has a small polar amino acid - serine - at −4, and binds preferentially to BRCT2 (*Figure 3E*) when presented in the context of a short peptide. However, the unfavourability of the small polar serine, which unlike arginine or glutamate could at least be sterically accommodated by BRCT1, is diminished in the context of a longer peptide (*Boos et al., 2011*). Binding to BRCT2 appears to be less selective beyond the requirement for a hydrophobic residue at −3. Thus, the Crb2-pT187 sequence is able to bind to Rad4-BRCT2 as well as Rad4-BRCT1 with comparable affinity, as can the Treslin-pS1001 sequence to TOPBP1-BRCT1 and TOPBP1-BRCT2. However, the ability to bind to BRCT1 does not guarantee binding to BRCT2, which may depend on the identity of the hydrophobic −3 residue. The RAD9-pS387 sequence for example, binds with low micromolar affinity to TOPBP1-BRCT1 where its −3 alanine and −4 leucine can effectively occupy the extended hydrophobic recess, but binds much less tightly to BRCT2 where only the minimal hydrophobic side chain of the −3 alanine is available to bind.

Many (but by no means all) of the sites shown to bind to TOPBP1-BRCT0,1,2 or Rad4-BRCT1,2 have a proline residue (at +1) following the phosphorylated serine or threonine, and are known (or presumed to be) targets for phosphorylation by CDKs or other proline-directed protein kinases (*Boos et al., 2011*; *Qu et al., 2013*). In the observed binding modes for both BRCT1 and BRCT2, the +1 residue is directed away from the body of the protein, and makes no direct contribution to the binding specificity, but is nonetheless accommodated.

## Identification of the TOPBP1-BRCT1,2 binding site in RHNO1

Based on the crystal structures described above, and the model for specificity derived from them, we set out to identify hitherto unrecognised interaction motifs in proteins implicated by genetic or proteomic studies in interaction with TOPBP1, but where the basis for that interaction has not been characterised.

RHNO1 (aka RHINO, RAD9-Hus1-Rad1 Interacting Nuclear Orphan) was identified as a contributor to DNA damage checkpoint signalling, which physically couples the 9-1-1 (RAD9-HUS1-RAD1) checkpoint clamp to TOPBP1 (*Cotta-Ramusino et al., 2011*; *Lindsey-Boltz et al., 2015*). While deletion analysis implicated the N-terminal half of RHNO1 in 9-1-1 binding, the location of the interaction with TOPBP1 was not defined. Using the TOPBP1-binding motifs defined above, a search of the human RHNO1 sequence using ScanProsite (*de Castro et al., 2006*) identified a good match in the C-terminal region of RHNO1 centred on Thr202 (*Figure 4A*). Although not so far annotated in phosphorylation site databases, this sequence has the characteristics of a proline-directed kinase site such as those identified in other TOPBP1/Rad4-binding sequences, and with a triplet of hydrophobic residues in the −5, −4 and −3 positions that would facilitate high-affinity binding to BRCT1 according to our model. The key attributes of this motif are highly conserved in RHNO1 homologues in metazoa (*Figure 4—figure supplement 1*).

To test the hypothesis that this is a *bona fide* TOPBP1-binding site, we synthesised a phosphopeptide encapsulating the putative RHNO1-pT202 site and measured its binding to segments of TOPBP1 using fluorescence polarisation. RHNO1-pT202 bound to TOPBP1-BRCT0,1,2 with sub-

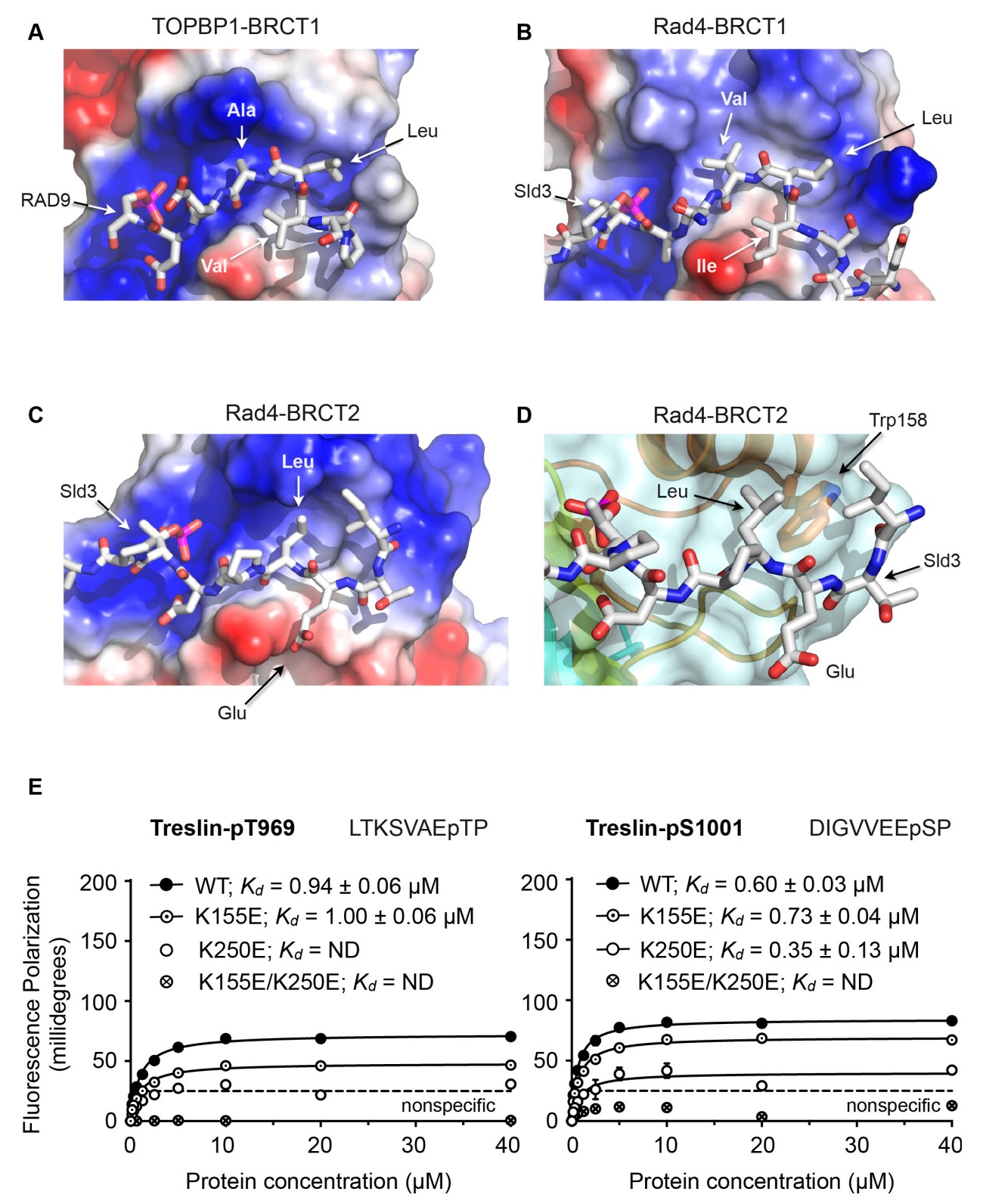

**Figure 3.** Structural basis of BRCT domain conformational preferences  (**A**) Detail of the RAD9-pSer387 peptide bound to TOPBP1-BRCT1, showing the tight turn conformation adopted by the consecutive hydrophobic residues at positions −5, −4 and −3 relative to the phosphorylated residue, that positions the side chains of the −4 and −3 residues in a pocket on the BRCT domain. (**B**) As A. but for the Sld3-pT650 peptide bound to Rad4-BRCT1. (**C**) Detail of the Sld3-pT650 peptide bound to Rad4-BRCT2. Unlike the BRCT1 ligand phosphopeptides, residues −3 and −4 have an extended main

*Figure 3 continued on next page*

*Figure 3 continued*

chain conformation, so that only the residue at −3 interacts with the hydrophobic pocket on the BRCT domain, so that the −4 position can accommodate a large polar amino acid such as glutamic acid. (D) The pocket in BRCT2 is constricted relative to that in BRCT1, by the presence of a tryptophan residue, which is topologically conserved in BRCT2 domains but absent from BRCT1. (E) Fluorescence polarisation assay of Treslin-derived phosphopeptides binding to the TOPBP1-BRCT0,1,2 segment. The Treslin-pT969 peptide, which has a large hydrophilic residue at −5 binds preferentially to BRCT2, whereas the Treslin-p1001 peptide with a glycine at −5 binds with comparable affinity to either. Both Treslin phosphopeptides have hydrophobic residues at the −3 positions.

DOI: https://doi.org/10.7554/eLife.39979.004

The following source data is available for figure 3:

**Source data 1.** Fluorescence polarisation titration of Treslin binding to TOPBP1.
DOI: https://doi.org/10.7554/eLife.39979.005

micromolar affinity (*Figure 4B*), but did not show any measurable binding to the other phospho-peptide binding modules of TOPBP1, that is BRCT4,5 and BRCT7,8 (*Figure 4C,D*). A dephosphorylated version of the same peptide did not bind.

We previously showed that charge-reversal mutations in conserved lysine residues in the phosphate-binding sites of BRCT1 and BRCT2 of TOPBP1 and Rad4, selectively block binding of phospho-peptide ligands to the mutated domain (*Boos et al., 2011*; *Qu et al., 2013*; *Rappas et al., 2011*) and we used this to determine whether RHNO1-pT202 bound BRCT1 or BRCT2 preferentially. We found that a TOPBP1-BRCT1,2 construct with a K250E mutation that disrupts binding to BRCT2, bound the peptide with comparable affinity to the wild-type, whereas no binding was observed with a TOPBP1-BRCT1,2 construct with a K155E mutant, which disrupts BRCT1, confirming TOPBP1-BRCT1 as the major binding site for RHNO1-pT202 (*Figure 4B*). Interestingly, truncation mutations of RHNO1 that eliminate the region of the protein in which this putative phosphorylation site occurs, and which are therefore unlikely to be unable to bind TOPBP1, have been identified in a number of families with hereditary pancreatic cancer (*Smith et al., 2016*).

## Identification of a novel Rad4-BRCT1,2 binding site in Mdb1

Mbd1 is the *S. pombe* orthologue of the metazoan DNA damage mediator protein MDC1 (*Wei et al., 2014*) and plays important roles in the DNA damage response. MDC1 is believed to interact with TOPBP1 through a central region containing six repeats of a degenerate SDT motif constitutively phosphorylated by CK2, which was originally implicated in mediating MDC1 interaction with NBS1 (*Chapman and Jackson, 2008*; *Melander et al., 2008*). The putative interaction of this region with TOPBP1 is believed to be mediated by the BRCT4,5 module, but is of significantly lower affinity than that displayed by other biologically significant phospho-peptide interactions with TOPBP1/Rad4 BRCT domains (*Leung et al., 2013*). Furthermore, these repeats do not occur in fission yeast Mdb1 where only a single SDT motif is evident.

To determine whether the single SDT site in Mdb1 mediates an interaction with Rad4, we synthesised a phospho-peptide encapsulating the bis-phosphorylated Mdb1-p216,p218 SDT site and measured its binding to segments of Rad4 using fluorescence polarisation (*Figure 5A*). While the Mdb1-p216,p218 peptide bound with sub-micromolar affinity to *S. pombe* Nbs1, consistent with observations in the human system, no binding was observed to either the BRCT1,2 or BRCT3,4 modules of Rad4.

To try and identify other potential phosphorylation sites on Mdb1 that might mediate its interaction with Rad4, we searched the Mdb1 sequence using ScanProsite as before (*de Castro et al., 2006*) and identified a good match to the motifs defined above, centred at a documented site of phosphorylation (*Beltrao et al., 2012*; *Ullah et al., 2016*) on Thr113 (*Figure 5B*). The sequence around Mdb1-Thr113 matches the TP/SP proline-directed kinase consensus and hydrophobic −3 residue seen for many TOPBP1/Rad4 BRCT1 and/or BRCT2 interacting sites. However, based on the presence of a polar residue, threonine at −4 our model would suggest that this site would not display high-affinity for BRCT1.

To characterise the interaction of this putative site with Rad4, we synthesised a phosphopeptide encapsulating the Mbd1-pT113 site and measured its binding to segments of Rad4 using fluorescence polarisation as above. We found that the Mbd1-pT113 peptide bound to Rad4-BRCT1,2 with

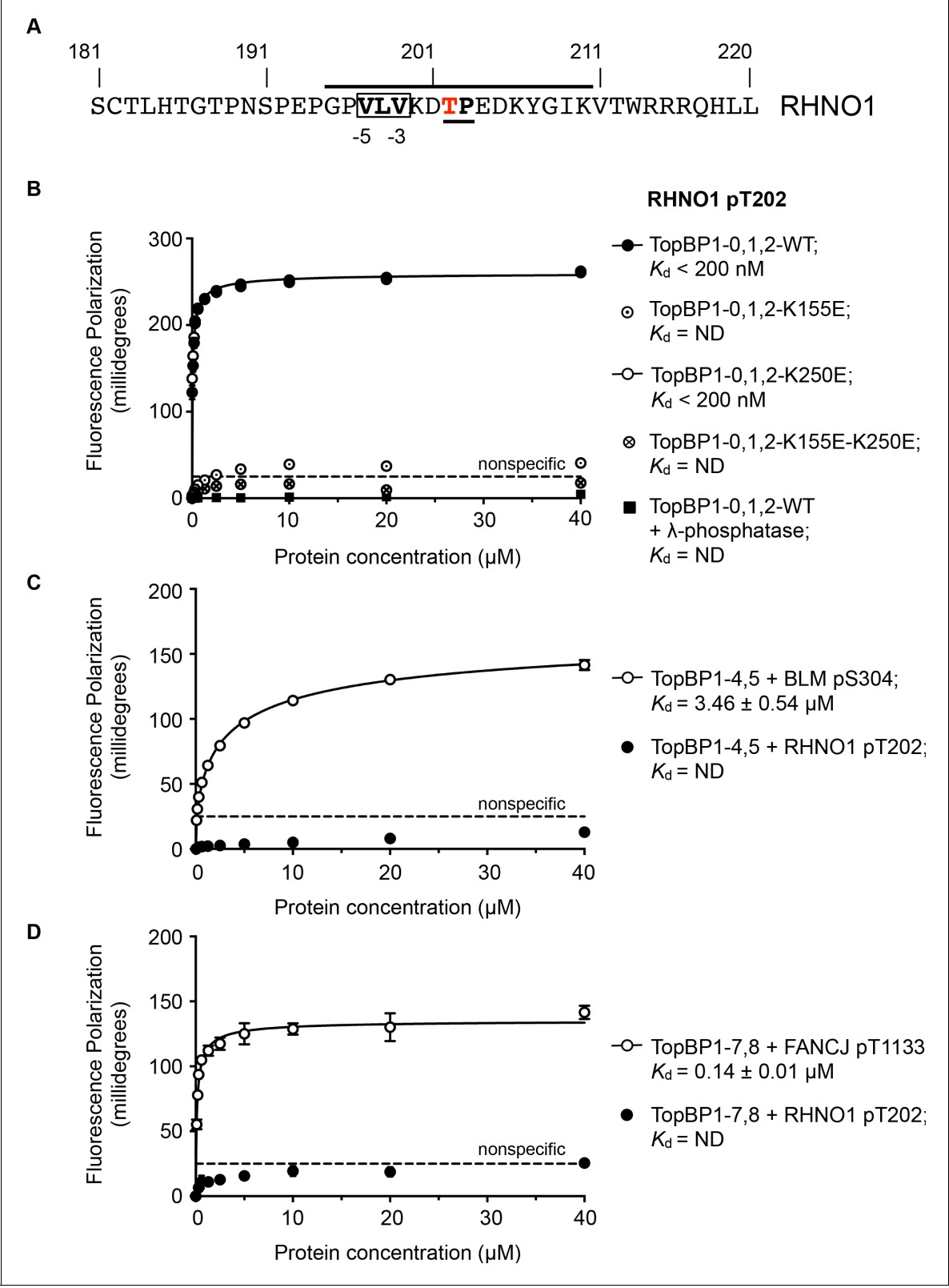

**Figure 4.** TOPBP1-binding site in RHNO1. (**A**) The C-terminal half of the 9-1-1 and TOPBP1-interacting scaffold protein RHNO1, contains a sequence motif with a potential phosphorylation site at Thr202, that corresponds closely to the consensus for preferential binding to Rad4/TOPBP1-BRCT1 or BRCT2 (**B**) Fluorescence polarisation assay of a RHNO1-derived phosphopeptide binding to the TOPBP1-BRCT0,1,2 segment. The RHNO1-pT202 peptide binds with high affinity to the wild-type BRCT0,1,2 construct, but fails to bind when dephosphorylated by λ-phosphatase. High affinity binding

*Figure 4 continued on next page*

*Figure 4 continued*

is lost in the presence of disruptive mutations in the phosphate-binding site of BRCT1, but is unaffected by comparable mutations in BRCT2. (**C**) In contrast to a documented phosphopeptide derived from BLM (*Blackford et al., 2015*), the RHNO1-pT202 peptide shows no affinity for the TOPBP1-BRCT4,5 segment. (**D**) As C – RHNO1-pT202 shows no affinity for the TOPBP1-BRCT7,8, unlike a documented phosphopeptide derived from FANCJ (*Gong et al., 2010*).

DOI: https://doi.org/10.7554/eLife.39979.006

The following source data and figure supplement are available for figure 4:

**Source data 1.** Fluorescence polarisation titration of RHNO1 binding to TOPBP1.
DOI: https://doi.org/10.7554/eLife.39979.008
**Source data 2.** Fluorescence polarisation titration of RHNO1 and BLM binding to TOPBP1.
DOI: https://doi.org/10.7554/eLife.39979.009
**Source data 3.** Fluorescence polarisation titration of RHNO1 and FANCJ binding to TOPBP1.
DOI: https://doi.org/10.7554/eLife.39979.010
**Figure supplement 1.** Conservation of the TOPBP1-interacting phosphorylation site in RHINO homologues.
DOI: https://doi.org/10.7554/eLife.39979.007

low micromolar affinity (*Figure 5C*) but showed no binding to Rad4-BRCT3,4 (*Figure 5—figure supplement 1*), the equivalent of TOPBP1-BRCT4,5 that has been implicated in mediating MDC1 interactions in the metazoan system (*Leung et al., 2013*). To determine which BRCT domains mediate the interaction with Mdb1-pT133, we measured binding to Rad4-BRCT1,2 with mutations in BRCT1 (K56E) or BRCT2 (K155E), and found binding was substantially reduced compared to wild-type with the BRCT2 mutant, but was largely unaffected by mutation of BRCT1 (*Figure 5C*).

Based on these observations, we were able to obtain a high-resolution crystal structure of Rad4-BRCT1,2 in complex with the Mdb1-pT113 peptide, confirming the preference for binding to BRCT2 (*Figure 5D*). The structure shows the Mdb1 peptide bound in a very similar conformation as Crb2-pT187, Crb2-pT235 and Sld3-pT650 peptides bound to this domain, with the valine at −3 relative to the phospho-threonine packed into the hydrophobic recess, while the polar threonine at −4 that makes the sequence incompatible with BRCT1, is directed out to solvent by the β-sheet conformation of the peptide backbone.

Deletion of Mdb1 only displays significant DNA damage phenotypes in the absence of some other Rad4-interacting proteins, and further work is required to determine whether the phosphorylation site we characterise here plays a critical role in Mdb1 function in vivo. However, analogous TOPBP1-binding sites in mammalian MDC1 appear to play a role in the response to DNA damage in mitotic cells (Ahorner, Jones et al., manuscript submitted).

## Discussion

As well as its eponymous interaction with DNA topoisomerase II (*Broderick et al., 2015*; *Yamane et al., 1997*), TOPBP1 and its homologues have been found to interact in a phosphorylation-dependent manner with a number of ligand proteins involved in different aspects of DNA replication and repair. These include, the CMG helicase assembly factor Treslin/Sld3 (*Boos et al., 2011*; *Kumagai et al., 2010*), the DNA damage mediators 53BP1/Crb2 and MDC1 (*Cescutti et al., 2010*; *Qu et al., 2013*; *Wang et al., 2011*), the 9-1-1 DNA damage checkpoint clamp (*Delacroix et al., 2007*; *Furuya et al., 2004*), and the DNA helicases BLM and FANCJ (*Blackford et al., 2015*; *Gong et al., 2010*). The data presented here add RHNO1 and the fission yeast homologue of MDC1, Mdb1, to this group, and there are no doubt further phosphorylation-dependent ligands yet to be identified.

While multiple interacting partners of TOPBP1 are known, the interplay between these has not been studied in depth, and whether multiple ligand proteins participate simultaneously with the same TOPBP1 scaffolded complex is unknown. Our data reveal clear and distinctive specificities for ligand binding between the two phosphopeptide-binding sites in the N-terminal BRCT module, and this is also likely to be true for the conserved central BRCT module, and for the C-terminal BRCT module, which is only present in metazoa. Apart from the contentious (see above) interaction claimed for BRCT5 with the SDT repeats of MDC1 (*Leung et al., 2013*), only two specific high-affinity interacting peptides have been identified for the central module – 406-DAFEGPpTQ-413 from

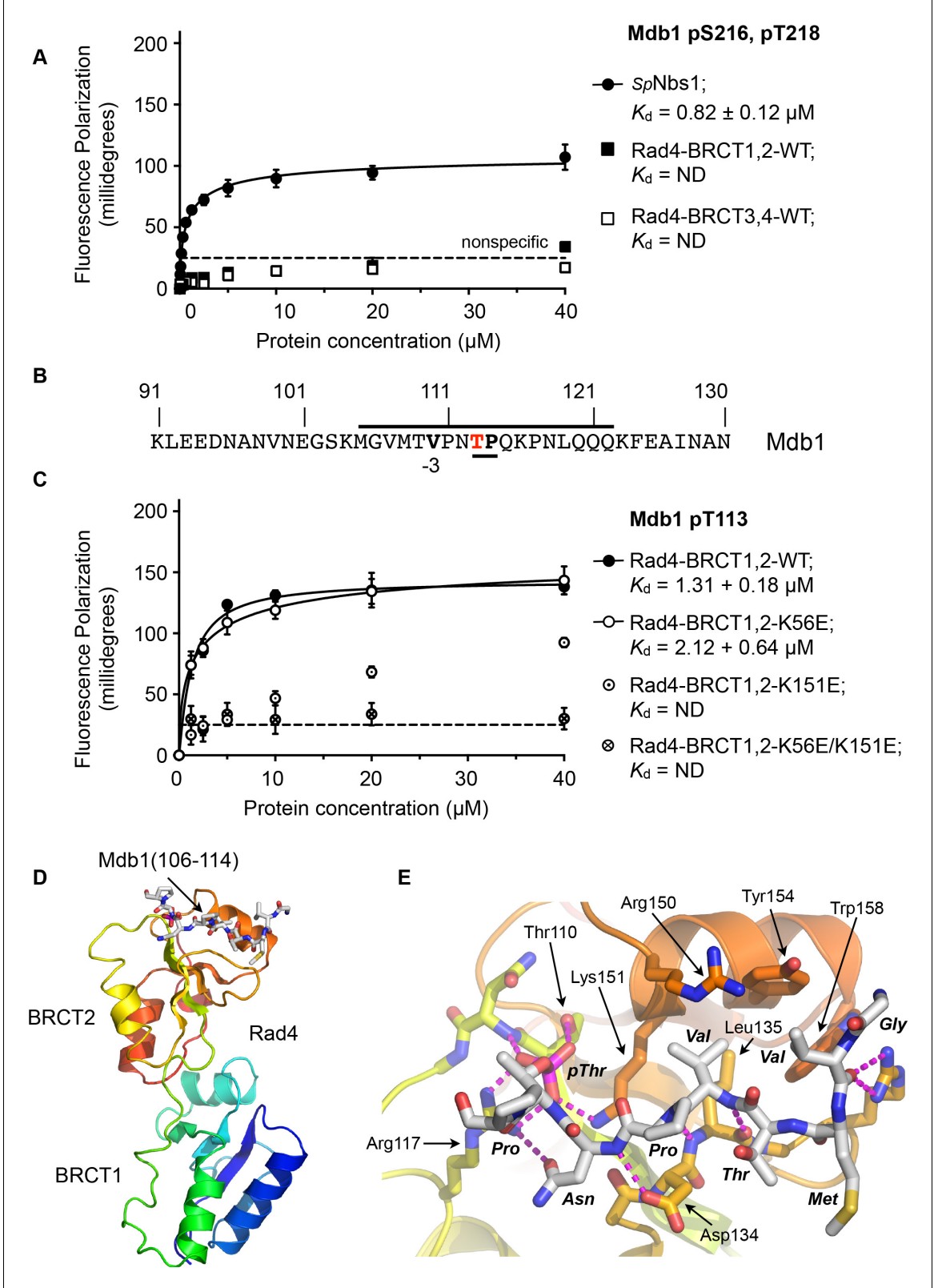

**Figure 5.** Rad4-binding site in Mdb1. (**A**) Fluorescence polarisation assay showing that a fluorescent peptide (Flu-GYGVEGDHS(pS)D(pT)EEEVVS) incorporating the single SDT site in Mdb1, the fission yeast homologue of MDC1, does not interact with Rad4-BRCT3,4 as has been suggested for the SDT sites in human MDC1 with the homologous TOPBP1-BRCT4,5 (***Leung et al., 2013***; ***Wang et al., 2011***). The Mdb1-SDT site peptide does, however, interact with high-affinity with the *S. pombe* homologue of NBS1. (**B**) Mdb1 contains a sequence motif with a potential phosphorylation site at

*Figure 5 continued on next page*

*Figure 5 continued*

Thr 113 that corresponds closely to the consensus for binding to Rad4/TOPBP1-BRCT2. (C) Fluorescence polarisation assay of an Mdb1-derived phosphopeptide binding to the Rad4-BRCT1,2 segment. The Mdb1-pT113 peptide binds with high affinity to the wild-type BRCT1,2 but only weakly when a disruptive mutation is introduced into the phosphate-binding site of BRCT2, but is largely unaffected by comparable mutations in BRCT1. All detectable bindings are lost when both BRCT domains are mutated. (D) Crystal structure of Mdb1-pT113 peptide bound to the BRCT2 domain within the Rad4-BRCT1,2 module. (E) Closeup of interactions. As in other complexes with BRCT2, the single hydrophobic residue at −3 relative to the phosphorylated residue binds into the hydrophobic pocket, with the main chain for the −2, –3 and −4 residues adopting an extended conformation, rather than the tight turn seen in interactions with BRCT1.

DOI: https://doi.org/10.7554/eLife.39979.011

The following source data and figure supplements are available for figure 5:

**Source data 1.** Fluorescence polarisation titration of Mdb1 STD sites binding to Rad4 or SpNbs1.

DOI: https://doi.org/10.7554/eLife.39979.014

**Source data 2.** Fluorescence polarisation titration of Mdb1 binding to Rad4.

DOI: https://doi.org/10.7554/eLife.39979.015

**Figure supplement 1.** Fluorescence polarisation assay of an Mdb1-derived phosphopeptide binding to the Rad4-BRCT3,4 segment.

DOI: https://doi.org/10.7554/eLife.39979.012

**Figure supplement 1—source data 1.** Fluorescence polarisation titration of Mdb1 binding to Rad4.

DOI: https://doi.org/10.7554/eLife.39979.013

the C-terminal tail of *S. pombe* Rad9, which binds to Rad4-BRCT4 (*Qu et al., 2013*), and 297-TDFVPPpSP-305 from BLM which binds to TOPBP1-BRCT5 (*Blackford et al., 2015*; *Sun et al., 2017*). These peptides both have a proline at −1 and a phenylalanine at −4, but further work will be required to determine the significance of these. Only a single high-affinity interaction has been validated for the canonical tandem BRCT7,8 module, with a peptide from FANCJ (aka BRIP/BACH1) containing pThr 1133 (*Leung et al., 2011*) (*Figure 4D*).

Based on these patterns of specificity, it is feasible that a single TOPBP1 molecule could, for example, simultaneously bind 9-1-1 (via BRCT1) and BLM (via BRCT5), or a single Rad4 molecule bind Crb2 (via BRCT1 and 2) and 9-1-1 (via BRCT4). Of course, steric constraints imposed by the rest of the ligand protein outside the phosphopeptide motif, might prevent simultaneous interaction, but multiple combinations are possible at least in principle.

Conversely, the mapped BRCT specificities of some ligands potentially rules out their co-existence within a TOPBP1/Rad4 scaffolded complex. Thus, both RHNO1 and the RAD9 component of the 9-1-1 clamp selectively bind BRCT1 of TOPBP1 and are therefore unlikely to be simultaneously bound to the same TOPBP1 molecule, in contradiction of current models (*Lindsey-Boltz et al., 2015*). Similarly, Treslin and 9-1-1 in metazoa, or Sld3 and Crb2 in fission yeast, would be in competition for binding to TOPBP1/Rad4. These competitions may be resolved by differences in affinity between binding motifs - for example RHNO1-pT202 appears to bind ~10 fold tighter than RAD9-pS387 – and by regulation of the phosphorylation state of individual ligand sites by cell cycle and/or DNA damage response systems. For example, while the RAD9-pS387 site is generated by CK2, which is constitutively active throughout the cell cycle (*Pinna, 2002*), the potentially competing RHNO1-pT202 site has the appearance of a CDK target, and its modification and interaction with TOPBP1 might be restricted to a specific cell cycle phase. Although these models for regulation of TOPBP1/Rad4 complex formation are appealing, the situation may be further complicated by the observation that under some circumstances TOPBP1 may be able to oligomerise via interaction of an AKT-dependent phosphorylation site on one TOPBP1 molecule, with the C-terminal BRCT7,8 module of another (*Liu et al., 2006*). Further work will be required to define the composition of the many different TOPBP1-scaffolded complexes that potentially occur within living cells.

Given the many roles of TOPBP1 in maintaining cell viability in the presence of the genomic instability that typically accompanies cancer progression, pharmacological disruption of TOPBP1-BRCT interactions with ligand proteins offers an attractive therapeutic approach. However, the poor cell-permeability of inhibitors targeting the strongly polar phosphate interactions on which binding to BRCT domains often depends, has limited progress (*Yuan et al., 2011*). The data presented here show that while phosphate interactions are important, specificity and high-affinity is mediated by a predominantly hydrophobic interaction that should be much more amenable to competitive blockade by cell-penetrant small molecules.

## Materials and methods

### Protein expression and purification

*E.coli* strain BL21(DE3) was transformed with modified pGEX-6P-1 or pET-15b vectors for expression of all protein constructs. Transformed bacteria were grown in a shaking incubator set at 37°C and 220 rpm in TurboBroth (Molecular Dimensions, Newmarket, UK) supplemented with the appropriate antibiotic for selection. Protein expression was induced by the addition of isopropyl β-D-1-thiogalactopyranoside (IPTG).

Cell pellets were resuspended in lysis buffer containing 50 mM HEPES pH 7.5, 200 mM NaCl and 0.25 mM TCEP supplemented with 50U Turbo DNase (ThermoFisher Scientific, Waltham, MA), disrupted by sonication, and the resulting lysate clarified by centrifugation at 40,000 x *g* for 60 min at 4°C.

For HIS and HIS-SUMO-tagged constructs the supernatant was applied to a 5 ml HiTrap TALON crude column (GE Healthcare, Little Chalfont, UK), washed first with buffer containing 50 mM HEPES pH 7.5, 1000 mM NaCl, 0.25 mM TCEP, and then again with buffer containing 50 mM HEPES pH 7.5, 200 mM NaCl, 0.25 mM TCEP, 10 mM imidazole. Retained protein was eluted by application of the same buffer supplemented with 250 mM imidazole.

For FP experiments, a Superdex 75 16/60 size exclusion column (GE Healthcare) was used to purify all proteins to >95% homogeneity in 25 mM HEPES pH 7.5, 150 mM NaCl, 1 mM EDTA, 0.25 mM TCEP, 0.02% (v/v) Tween-20.

For GST-tagged constructs, the supernatant was applied to a 5 ml HiTrap GST column (GE Healthcare), then washed with buffer containing 50 mM HEPES pH 7.5, 1000 mM NaCl, 0.25 mM TCEP. Retained protein was then eluted by application of the lysis buffer supplemented with 20 mM glutathione.

For FP experiments, a Superdex 200 16/60 size exclusion column (GE Healthcare) was used to purify all proteins to >95% homogeneity in 25 mM HEPES pH 7.5, 150 mM NaCl, 1 mM EDTA, 0.25 mM TCEP, 0.02% (v/v) Tween-20.

For crystallography experiments, affinity purification tags were removed by incubation with Human SENP1C or Human Rhinovirus 3C proteases to remove HIS-SUMO or GST tags respectively, before application of the recombinant protein to a Superdex 75 16/60 size exclusion column (GE Healthcare) equilibrated in buffer containing 10 mM HEPES pH 7.5, 100 mM NaCl, 0.5 mM TCEP.

### Peptide pull-down experiments method

Biotinylated *Hs*RAD9 peptides corresponding to the C-terminal 45 residues of RAD9 were synthesized with sequences as follows:

| pS387 | 347 – aepstvpgtpppkkfrslffgsilapvrspqgpspvlaed**ps**egeg – 391 |
|---|---|
| S387 | 347 - aepstvpgtpppkkfrslffgsilapvrspqgpspvlaed**s**egeg – 391 |
| S-1 | 347 – aepstvpgtpppkkfrslffgsilapvrspqgpspvlae**Sps**egeg – 391 |
| K-1 | 347 – aepstvpgtpppkkfrslffgsilapvrspqgpspvlae**Kps**egeg – 391 |
| L-1 | 347 – aepstvpgtpppkkfrslffgsilapvrspqgpspvlae**Lps**egeg – 391 |
| S-3 | 347 – aepstvpgtpppkkfrslffgsilapvrspqgpspvl**Sedps**egeg – 391 |
| F-3 | 347 – aepstvpgtpppkkfrslffgsilapvrspqgpspvl**Fedps**egeg – 391 |
| D-3 | 347 – aepstvpgtpppkkfrslffgsilapvrspqgpspvl**Dedps**egeg – 391 |
| G-3 | 347 – aepstvpgtpppkkfrslffgsilapvrspqgpspvl**Gedps**egeg – 391 |
| I-3 | 347 – aepstvpgtpppkkfrslffgsilapvrspqgpspvl**Iedps**egeg – 391 |
| V-3 | 347 – aepstvpgtpppkkfrslffgsilapvrspqgpspvl**Vedps**egeg – 391 |
| S-4 | 347 – aepstvpgtpppkkfrslffgsilapvrspqgpspv**Saedps**egeg – 391 |
| N-4 | 347 – aepstvpgtpppkkfrslffgsilapvrspqgpspv**Naedps**egeg – 391 |
| A-4 | 347 – aepstvpgtpppkkfrslffgsilapvrspqgpspv**Aaedps**egeg – 391 |
| F-4 | 347 – aepstvpgtpppkkfrslffgsilapvrspqgpspv**Faedps**egeg – 391 |

*Continued on next page*

| I-4 | 347 – aepstvpgtpppkkfrslffgsilapvrspqgpsp**vl**aed**ps**egeg – 391 |
|-----|------------------------------------------------------------------|
| Q-5 | 347 – aepstvpgtpppkkfrslffgsilapvrspqgpsp**Q**laed**ps**egeg – 391 |
| S-5 | 347 – aepstvpgtpppkkfrslffgsilapvrspqgpsp**S**laed**ps**egeg – 391 |
| D-5 | 347 – aepstvpgtpppkkfrslffgsilapvrspqgpsp**D**laed**ps**egeg – 391 |
| F-5 | 347 – aepstvpgtpppkkfrslffgsilapvrspqgpsp**F**laed**ps**egeg – 391 |
| A-5 | 347 – aepstvpgtpppkkfrslffgsilapvrspqgpsp**A**laed**ps**egeg – 391 |

For pull-down experiments, 10 µl streptavidin agarose were saturated with a given peptide in binding buffer (20 mM HEPES 8.0, 450 mM NaCl, 0.01% Triton, 5% Glycerol, 5 mM MgCl2, 5 mM 2-mercaptoethanol). N-Myc6-TEV2-hTOPBP1-1-360 in vitro translated in rabbit reticulocyte lysate in the presence of $^{35}$S-methionine in binding buffer was added and incubated for 2 hr at 4°C. After washing in binding buffer, inputs and beads were analysed by SDS PAGE and autoradiography.

## Bioinformatics and peptide identification

Consensus motifs based on the observed crystal structures and binding data for TOPBP1/Rad4 BRCT1 and BRCT2 domains, were compiled as Prosite strings, as follows,

BRCT1: [FILMV]-[FILMV]-[FILMV]-x-x-[ST]-P, BRCT2: [FILMV]-x-x-[ST]-P and scanned over the amino acid sequences of putative TOPBP1/Rad4-interacting proteins using the ExPASy ScanProsite server. Putative interactors were identified from the literature and from the DNA Damage Response interaction network compiled in *Pearl et al. (2015)*. Matches to the motifs were prioritised for experimental analysis on the basis of occurrence in non-structured regions, annotation as phosphorylated, and conservation across species. As well as the sites in human RHNO1 and *S. pombe* Mbd1 reported here, potential TOPBP1-binding motifs were also identified in 53BP1, MDC1 and SMARCAD1.

## Fluorescence polarisation experiments

Fluorescein-labelled peptides (Peptide Protein Research Ltd, Fareham, UK) at a concentration of 200 nM, were incubated at room temperature with increasing concentrations of recombinant protein in 25 mM HEPES pH 7.5, 150 mM NaCl, 1 mM EDTA, 0.25 mM TCEP, 0.02% (v/v) Tween-20. Four parallel replicates of each titration were dispensed into a black 96-well polypropylene plate (VWR, Lutterworth, UK) for measurement. Fluorescence polarisation was measured in a CLARIOstar multimode microplate reader (BMG Labtech, Aylesbury, UK). Binding curves represent the mean of 4 independent experiments, with error bars indicating one standard deviation. All data were fitted by non-linear regression, to a one site-specific binding model in Prism seven for Mac OS X (v 7.0d, GraphPad Software) in order to calculate the reported dissociation constants ($K_d$). Low-affinity non-specific responses for which a binding curve could not be determined are marked as ND.

Fluorescent Phosphopeptides used:
RHNO1 pT202 Fluorescein-GYGGPVLVKDpTPEDKYGI
Treslin pT969 Fluorescein-GYGLTKSVAEpTPVHKQIS
Treslin pT1001 Fluorescein-GYGDIGVVEEpSPEKGDEI
BLM pS304 Fluorescein-GYGDTDFVPPpSPEEIISA
FANCJ pT1133 Fluorescein-GYGEDESIYFpTPELYDPE
Mdb1 pT113 Fluorescein-GYGGVMTVPNpTPQKPNLQ
Mdb1 pS216, pT218 Fluorescein-GYGVEGDHSpSDpTEEEVVS

## Crystallisation, data collection, phasing, model building rnd Refinement

Co-crystallisation trials of cTOPBP1-BRCT0,1,2 mixed with a two-fold excess of RAD9 peptide (SPVLAEDpSEGE) were set up in MRC 2 96-well sitting-drop vapour-diffusion plates by mixing 200 nl of 10 mg/ml protein solution with 200 nl of mother liquor, over a well volume of 50 µl. Crystals that grew from condition Morpheus E9 (100 mM bicine/Trizma base pH 8.5, 300 mM diethyleneglycol, 300 mM triethyleneglycol, 300 mM tetraethyleneglycol, 300 mM pentaethyleneglycol, 10% w/v PEG 20000% and 20% v/v PEG MME 550) were harvested and plunged into liquid nitrogen.

Co-crystallisation trials of *Sp*Rad4-BRCT1,2 mixed with a two-fold excess of *Sp*Sld3 pT636 pT650 peptide (GYDSILVQApTPRKSSSVITELPDpTPIKMNS) were set up in MRC 2 96-well sitting-drop

vapour-diffusion plates by mixing 200 nl of 12 mg/ml protein solution with 200 nl of mother liquor, over a well volume of 50 μl.

Co-crystallisation trials of SpRad4-BRCT1,2 mixed with a two-fold excess of SpMdb1 pT113 peptide (GVMTVPNpTPQKPNLQ) were set up in MRC 2 96-well sitting-drop vapour-diffusion plates by mixing 200 nl of 12 mg/ml protein solution with 200 nl of mother liquor, over a well volume of 50 μl. Crystals that grew from condition STRUCTURE D1 (200 mM Sodium acetate trihydrate, 100 mM Tris 8.5% and 30 % w/v PEG 4000) were soaked in mother liquor with the addition of 25% (v/v) glycerol prior to plunging in liquid nitrogen.

For the Rad4 BRCTs 1,2 - Mdb1 pT113, and TOPBP1 BRCTs 0,1,2 – RAD9 pS387 structures, diffraction data were collected at the I04 beamline, while for Rad4 BRCTs 1,2 - Sld3 pT636 pT650 data were collected at the I02 beamline, at the Diamond Synchrotron Lightsource (Didcot, UK) and all data were processed using the Xia2 software pipeline (*Winter et al., 2013*).

Processing and refinement were carried out using the CCP4 and PHENIX suites of programmes. The peptide bound structures were phased by molecular replacement using Phaser (*McCoy, 2007*) and PDB entries 4BMC for the two Rad4 structures and 2XNH for the TOPBP1 structure as search models. The structures were built using COOT (*Emsley et al., 2010*) and refined with Phenix (*Adams et al., 2011*) or REFMAC5 (*Murshudov et al., 2011*). Statistics for data collection and structure refinement are provided in *Supplementary file 1*. Coordinates and structure factors have been deposited in the Protein Databank with accession codes: 6HM5 (TOPBP1-RAD9 complex), 6HM4 (Rad4-Mdb1 complex), 6HM3 (Rad4-Sld3 complex).

## Acknowledgements

We thank Mark Roe for assistance with X-ray data collection, and John Diffley and Tony Carr for helpful discussion and provision of reagents. We are grateful to the Diamond Light Source Ltd., Didcot, UK, for access to synchrotron radiation and to the Wellcome Trust for their support of X-ray diffraction facilities at the University of Sussex. This work was supported by Cancer Research UK Programme Grants C302/A14532 and C302/A24386 (AWO and LHP).

## Additional information

### Funding

| Funder | Grant reference number | Author |
| --- | --- | --- |
| Cancer Research UK | C302/A14532 | Laurence H Pearl<br>Matthew Day<br>Antony Oliver |
| Cancer Research UK | C302/A24386 | Laurence H Pearl<br>Matthew Day<br>Antony Oliver |

The funders had no role in study design, data collection and interpretation, or the decision to submit the work for publication.

### Author contributions

Matthew Day, Conceptualization, Validation, Visualization, Methodology, Writing—review and editing; Mathieu Rappas, Katie Ptasinska, Investigation; Dominik Boos, Investigation, Writing - review and editing; Antony W Oliver, Conceptualization, Supervision, Funding acquisition, Validation, Visualization, Investigation, Methodology, Project administration, Writing—review and editing; Laurence H Pearl, Conceptualization, Supervision, Funding acquisition, Validation, Visualization, Methodology, Project administration, Writing - original draft, Writing—review and editing

### Author ORCIDs

Antony W Oliver https://orcid.org/0000-0002-2912-8273
Laurence H Pearl https://orcid.org/0000-0002-6910-1809

Decision letter and Author response
Decision letter https://doi.org/10.7554/eLife.39979.025
Author response https://doi.org/10.7554/eLife.39979.026

## Additional files

### Supplementary files

• Supplementary file 1. Crystallographic data collection and refinement statistics.
DOI: https://doi.org/10.7554/eLife.39979.017

• Transparent reporting form
DOI: https://doi.org/10.7554/eLife.39979.018

### Data availability

Coordinates and structure factors have been deposited in the Protein Databank with accession codes: 6HM5 (TOPBP1-RAD9 complex), 6HM4 (Rad4-Mdb1 complex) and 6HM3 (Rad4-Sld3 complex).

The following datasets were generated:

| Author(s) | Year | Dataset title | Dataset URL | Database, license, and accessibility information |
|---|---|---|---|---|
| Matthew Day, Antony Oliver, Laurence H Pearl | 2018 | Crystal structure of TOPBP1 BRCT0,1,2 in complex with a RAD9 phosphopeptide | https://www.rcsb.org/structure/6HM5 | 6HM5 |
| Matthew Day, Antony Oliver, Laurence H Pearl | 2018 | Crystal structure of Rad4 BRCT1,2 in complex with a Mdb1 phosphopeptide | https://www.rcsb.org/structure/6HM4 | 6HM4 |
| Mathieu Rappas, Matthew Day, Antony Oliver, Laurence H Pearl | 2018 | Crystal structure of Rad4 BRCT1,2 in complex with a Sld3 phosphopeptide | https://www.rcsb.org/structure/6HM3 | 6HM3 |

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
