## [Decision Letter]

[**Editorial note:** This article has been through an editorial process in which the authors decide how to respond to the issues raised during peer review. The Reviewing Editor's assessment is that all the issues have been addressed.]

Thank you for submitting your article "BRCT domains of the DNA damage checkpoint proteins TOPBP1/Rad4 display distinct specificities for phosphopeptide ligands" for consideration by *eLife*. Your article has been reviewed by three peer reviewers, including Volker Dötsch as the Reviewing Editor and Reviewer #1, and the evaluation has been overseen by John Kuriyan as the Senior Editor. The other reviewers remain anonymous.

The Reviewing Editor has highlighted the concerns that require revision and/or responses, and we have included the separate reviews below for your consideration. If you have any questions, please do not hesitate to contact us.

Summary:

Day et al. describe a detailed interaction study of the BRCT domains in mammalian TOPBP1 and in the fission yeast homologue RAD4 with phosphopeptides. They analyze by structure determination of domain – peptide complexes the sequence requirement for strong interactions. Using these structures they show that residues in the -3/-4 position are responsible for selective binding of peptides to BRCT1 vs. BRCT2. With this sequence information in hand they search for other potential binding partners and identify potential phoshopeptides in other proteins that might mediate interaction with TOPBP1/RAD4 upon phosphorylation.

The data are of high quality and the results provide an important addition to understand selective interaction / signalling events and will be very important for researchers working on understanding selective phosphopeptide interactions.

Major concerns:

1) The search criteria for identifying the peptides in RHNO1 and Mdb1 are not clear and should be better defined. Using the information provided by the authors more peptide sequences than described in the manuscript would be possible.

2) It should be discussed if consensus sequences have been identified for any other BRCT domains.

3) The paper would be substantially strengthened by validation of the novel-binding sites in cells by confirming the interaction between TOPBP1 and RHNO1 with pull down experiments and mutants such as RHNO1 T202A or VLV.

4) The figure legends should be redesigned using color for better visualization of the results.

Title: The title of this manuscript emphasizes that BRCT domains "display distinct specificities", which seems to be a point that has been shown before. A title like "Structural basis of phosphopeptide ligand binding specificity of BRCT domains of TOPBP1/Rad4" may better reflect the main findings of this paper.

Separate reviews (please respond to each point):

*Reviewer #1:*

Day et al. describe a detailed interaction study of the BRCT domains in mammalian TOPBP1 and in the fission yeast homologue RAD4 with phosphopeptides. They analyze by structure determination of domain – peptide complexes the sequence requirement for strong interactions. Using these structures they show that residues in the -3/-4 position are responsible for selective binding of peptides to BRCT1 vs. BRCT2. With this sequence information in hand they search for other potential binding partners and identify potential phoshopeptides in other proteins that might mediate interaction with TOPBP1/RAD4 upon phosphorylation.

The data are of high quality and the results provide an important addition to understand selective interaction / signalling events.

However, the figure legend within the Figures 3, 4 and 5 should be presented in a clear understandable color code.

*Reviewer #2:*

In this work, the authors applied structural biology analysis to determine the mechanistic basis of phosphopeptide ligand binding specificity of two BRCT domains (BRCT1 and BRCT2) in TOPBP1 and Rad4, and revealed distinct amino acid sequence characteristics of high affinity ligands for BRCT1 and BRCT2. Furthermore, they showed that their findings have predictive power by identifying a TOPBP1-binding site in RHNO1 and a Rad4-binding site in Mdb1. These results shed important new light on the ligand recognition mechanism of BRCT domains and are of high value to researchers working on BRCT-domain-containing proteins and DNA damage response.

Major comments:

1) The identifications of TOPBP1/Rad4-binding sites in RHNO1 and Mdb1 are highlights of this paper. However, the authors failed to describe in sufficient detail how these sites were found. In the main text, they only mentioned that ScanProsite software was used for the search of sequences matching the consensus motif. I think the authors should add a section in the Materials and methods part to describe which consensus motif sequences were used for the search and how many matching sequences were found in RHNO1 and Mdb1. For the BRCT2 ligand, besides the phosphorylatable residue, only -3 position has a requirement of being hydrophobic. Such a sequence pattern should occur many times in a protein. For example, in Mdb1, even adding the requirement of +1 position being Pro, there are two other sites besides T113 that can match the consensus: S426 (421-FVILGSP-427) and T85 (80-VRINDTP-86). I suspect that other criteria need to be applied to further filtering the candidate sites. In the case of RHNO1, the authors showed that sequence conservation can be used as a further support. This criterion should also work for Mdb1, as the T113 site is conserved among the fission yeast species.

2) I think it would be useful to the readers if the authors can discuss whether any sequence consensus have been found for the ligands of other BRCT domains in TOPBP1/Rad4 and BRCT domains in other proteins, and whether the insights gained in this study on BRCT1 and BRCT2 can be applied to other BRCT domains.

Minor comments:

1) "*S.pombe*" in the Abstract and elsewhere should be "*S. pombe*", i.e. with a space between "*S.*" and "*pombe*".

2) In the peptide pull-down assay shown in Figure 1D, the -1 position was mutated but the author did not describe the result in the main text.

3) Figure 3C. There is a mistake in the figure legend. The peptide shown in this panel should be the Sld3-pT650 peptide, not the Sld3-pT636 peptide.

4) Figure 3E. Figure 4B-D, Figure 5A and 5C, circles and squares are not shown properly in the figure keys.

5) Figure 3E legend says "The Treslin-pT969 peptide, which has a small hydrophilic residue at -5 binds preferentially to BRCT2". The -5 residue is K, which is not a small hydrophilic residue.

6) Figure 3E legend says "Both Treslin phosphopeptides have hydrophobic residues at -3 and -4 positions." This is not correct, as the -4 residue in the Treslin-pT969 peptide is S, not a hydrophobic residue.

7) In Introduction, the sentence "genetic and biochemical studies show that they are nonetheless selective and specific for the phosphopeptide sequences with which they interact" should have references associated with it. I suspect that the authors can move the references before this sentence to after this sentence.

8) The title of this manuscript emphasizes that BRCT domains "display distinct specificities", which seems to be a point that has been shown before (see the sentence in Introduction quoted in minor comment 7). I think a title like "Structural basis of phosphopeptide ligand binding specificity of BRCT domains of TOPBP1/Rad4" may better reflect the main findings of this paper.

*Reviewer #3:*

Day et al. presented crystal structures of TOPBP1/Rad4 BRCT0,1,2 domain bound to phospho-peptides from RAD9 and Sld3 and compared the binding modes with existing structures of Rad4 BRCT1,2 domain bound to phospho-peptides to define the determinants of BRCT1 and BRCT2 domain specificity. They found that most peptides contain a preferred phospho-TP/SP motif for proline-directed kinase. Peptides that bind to BRCT1 undergo a tight turn conformation at -3 to -4 position and harbour hydrophobic residues at -3 to -5 position. Surprisingly BRCT2 contains a conserved tryptophan that precludes the tight turn conformation and seems to be less selective at -4 and -5 position as long as -3 is a hydrophobic residue with the exception of an alanine. Using this information, the authors identified novel TOPBP1-BRCT1 binding site in RHNO1 and Rad4-BRCT2 binding site in Mdb1 and validated the binding mode by fluorescence polarisation assays and a crystal structure of Rad4 BRCT1,2 in complex with Mdb-pT113 peptide.

Strength: The data are solid and the manuscript is well written. The findings are interesting as it defined the rules for TOPBP1/Rad4 BRCT1 and BRCT2 domain phospho-peptide specificity. This should facilitate future studies directed at identification of other phospho-ligands and understanding of how TOPBP1/Rad4 engage multiple binding partners.

Weakness: Lack of validation of the novel-binding site in cells. It would strengthen the hypothesis by confirming the interaction between TOPBP1 and RHNO1 in cells with pull down experiments using RHNO1 mutants such as T202A or VLV mutant.

Minor comments:

1) It would be useful to mention how K155E and K250E disrupt the BRCT1 and BRCT2 phospho-peptide binding site, respectively, when discussing Treslin in the second paragraph of the subsection “Structural basis for peptide selectivity by BRCT1 and BRCT2” rather than later in the last paragraph of the subsection “Identification of the TOPBP1-BRCT1,2 binding site in RHNO1”, for clarity.

2) Please show the Mdb1 pS216, pT218 peptide sequence in Figure 5. Perhaps the sequence is obvious that it doesn't bind BRCT1,2 based on the prediction.

3) The symbol legends for all fluorescence polarisation curves are not displayed correctly. Please specify what ND stands for.

4) In Figure 5C, please explain "inaccurate" and how 20-fold reduction (subsection “Identification of a novel Rad4-BRCT1,2 binding site in Mdb1”, fourth paragraph) was derived when comparing the Kd of BRCT1,2 WT and K151E mutant for Mdb1 pT113 peptide.

---

## [Author Response]

Reviewer #1:

Day et al. describe a detailed interaction study of the BRCT domains in mammalian TOPBP1 and in the fission yeast homologue RAD4 with phosphopeptides. They analyze by structure determination of domain – peptide complexes the sequence requirement for strong interactions. Using these structures they show that residues in the -3/-4 position are responsible for selective binding of peptides to BRCT1 vs. BRCT2. With this sequence information in hand they search for other potential binding partners and identify potential phoshopeptides in other proteins that might mediate interaction with TOPBP1/RAD4 upon phosphorylation.The data are of high quality and the results provide an important addition to understand selective interaction / signalling events.However, the figure legend within the Figures 3, 4 and 5 should be presented in a clear understandable color code.

The confusion over the figure legends on Figures 3, 4, 5 was due to a PDF conversion problem moving from Keynote and Adobe Illustrator in which the figures were prepared. This resulted in parts of the plotting symbols being lost, so that several of the curves were just referenced as open circles in the legend. This has been repaired and now works fine in black line, as was originally intended.

Reviewer #2:

In this work, the authors applied structural biology analysis to determine the mechanistic basis of phosphopeptide ligand binding specificity of two BRCT domains (BRCT1 and BRCT2) in TOPBP1 and Rad4, and revealed distinct amino acid sequence characteristics of high affinity ligands for BRCT1 and BRCT2. Furthermore, they showed that their findings have predictive power by identifying a TOPBP1-binding site in RHNO1 and a Rad4-binding site in Mdb1. These results shed important new light on the ligand recognition mechanism of BRCT domains and are of high value to researchers working on BRCT-domain-containing proteins and DNA damage response.Major comments:1) The identifications of TOPBP1/Rad4-binding sites in RHNO1 and Mdb1 are highlights of this paper. However, the authors failed to describe in sufficient detail how these sites were found. In the main text, they only mentioned that ScanProsite software was used for the search of sequences matching the consensus motif. I think the authors should add a section in the Materials and methods part to describe which consensus motif sequences were used for the search and how many matching sequences were found in RHNO1 and Mdb1. For the BRCT2 ligand, besides the phosphorylatable residue, only -3 position has a requirement of being hydrophobic. Such a sequence pattern should occur many times in a protein. For example, in Mdb1, even adding the requirement of +1 position being Pro, there are two other sites besides T113 that can match the consensus: S426 (421-FVILGSP-427) and T85 (80-VRINDTP-86). I suspect that other criteria need to be applied to further filtering the candidate sites. In the case of RHNO1, the authors showed that sequence conservation can be used as a further support. This criterion should also work for Mdb1, as the T113 site is conserved among the fission yeast species.

The reviewer’s comments are well taken and we have included a section in the Materials and methods as requested. Conservation is a useful consideration, but as we comment in the revised manuscript, the location of the motif outside a structured segment of the protein is also important. The two other matches the reviewer notes in Mdb1 T85 and S426, both occur in secondary structural elements within globular FHA and BRCT domains, respectively, which would prevent their interaction with Rad4 in the conformation required without substantial unfolding, and loss of functions of those domains.

2) I think it would be useful to the readers if the authors can discuss whether any sequence consensus have been found for the ligands of other BRCT domains in TOPBP1/Rad4 and BRCT domains in other proteins, and whether the insights gained in this study on BRCT1 and BRCT2 can be applied to other BRCT domains.

This is a very useful suggestion, and we have included a discussion of what little is known for the BRCT4,5 and BRCT7,8 modules in the revised manuscript.

Minor comments:1) "S.pombe" in the Abstract and elsewhere should be "S. pombe", i.e. with a space between "S." and "pombe".

Done.

2) In the peptide pull-down assay shown in Figure 1D, the -1 position was mutated but the author did not describe the result in the main text.

Added.

3) Figure 3C. There is a mistake in the figure legend. The peptide shown in this panel should be the Sld3-pT650 peptide, not the Sld3-pT636 peptide.

Corrected.

4) Figure 3E. Figure 4B-D, Figure 5A and 5C, circles and squares are not shown properly in the figure keys.

PDF conversion error – rectified – see above.

5) Figure 3E legend says "The Treslin-pT969 peptide, which has a small hydrophilic residue at -5 binds preferentially to BRCT2". The -5 residue is K, which is not a small hydrophilic residue.

Corrected.

6) Figure 3E legend says "Both Treslin phosphopeptides have hydrophobic residues at -3 and -4 positions." This is not correct, as the -4 residue in the Treslin-pT969 peptide is S, not a hydrophobic residue.

Corrected.

7) In Introduction, the sentence "genetic and biochemical studies show that they are nonetheless selective and specific for the phosphopeptide sequences with which they interact" should have references associated with it. I suspect that the authors can move the references before this sentence to after this sentence.

Done.

8) The title of this manuscript emphasizes that BRCT domains "display distinct specificities", which seems to be a point that has been shown before (see the sentence in Introduction quoted in minor comment 7). I think a title like "Structural basis of phosphopeptide ligand binding specificity of BRCT domains of TOPBP1/Rad4" may better reflect the main findings of this paper.

We would prefer to keep the title as it is.

Reviewer #3:

Day et al. presented crystal structures of TOPBP1/Rad4 BRCT0,1,2 domain bound to phospho-peptides from RAD9 and Sld3 and compared the binding modes with existing structures of Rad4 BRCT1,2 domain bound to phospho-peptides to define the determinants of BRCT1 and BRCT2 domain specificity. They found that most peptides contain a preferred phospho-TP/SP motif for proline-directed kinase. Peptides that bind to BRCT1 undergo a tight turn conformation at -3 to -4 position and harbour hydrophobic residues at -3 to -5 position. Surprisingly BRCT2 contains a conserved tryptophan that precludes the tight turn conformation and seems to be less selective at -4 and -5 position as long as -3 is a hydrophobic residue with the exception of an alanine. Using this information, the authors identified novel TOPBP1-BRCT1 binding site in RHNO1 and Rad4-BRCT2 binding site in Mdb1 and validated the binding mode by fluorescence polarisation assays and a crystal structure of Rad4 BRCT1,2 in complex with Mdb-pT113 peptide.Strength: The data are solid and the manuscript is well written. The findings are interesting as it defined the rules for TOPBP1/Rad4 BRCT1 and BRCT2 domain phospho-peptide specificity. This should facilitate future studies directed at identification of other phospho-ligands and understanding of how TOPBP1/Rad4 engage multiple binding partners.Weakness: Lack of validation of the novel-binding site in cells. It would strengthen the hypothesis by confirming the interaction between TOPBP1 and RHNO1 in cells with pull down experiments using RHNO1 mutants such as T202A or VLV mutant.

This point has not escaped our notice. However, this is not a straightforward binary interaction system in which the kind of simple pull-down experiment the reviewer suggests, will be informative. The current understanding of RHNO1, is that it interacts with TOPBP1 – most likely through the phosphorylation site we identify here – but it also interacts with the 9-1-1 checkpoint clamp, which itself interacts directly with TOPBP1 independently of RHNO1. Thus, TOPBP1, 9-1-1 and RHNO1 are apices of an interaction triangle in which pull-down of any apex will co-precipitate both other apices. Thus, even if a RHNO1 T202A mutation prevented RHNO1 phosphorylation in vivo and thereby disrupted the RHNO1-TOPBP1 ‘edge’ of the interaction triangle, as our in vitro data would predict, the interaction of RHNO1 with 9-1-1 would still allow co-precipitation of TOPBP1 through the high-affinity RAD9 – TOPBP1 interaction. As part of our ongoing studies of this system, we have now mapped the regions of RHNO1 that mediate its interaction with different components of 9-1-1 at least in vitro, and are developing an RHNO1 CRISPR/Cas9 knockout cell line into which RHNO1 constructs with defined mutations can be engineered and their interactions and functionality assessed. Along with a phosphospecific antibody to RHNO1-pThr202, which we are also developing, this will allow us to properly test the contribution of the pThr202 site to RHNO1 – TOPBP1 interaction, and fully characterise the role of RHNO1 in the 9-1-1 – TOPBP1 checkpoint system. Clearly this is a substantial piece of work in its own right that goes far beyond the scope of this current manuscript.

Minor comments:1) It would be useful to mention how K155E and K250E disrupt the BRCT1 and BRCT2 phospho-peptide binding site, respectively, when discussing Treslin in the second paragraph of the subsection “Structural basis for peptide selectivity by BRCT1 and BRCT2” rather than later in the last paragraph of the subsection “Identification of the TOPBP1-BRCT1,2 binding site in RHNO1”, for clarity.

We don’t really see a suitable place to include this as the reviewer suggests, but to clarify the purpose of these charge-reversal mutations, we have expanded the preamble to the experimental description in the last paragraph of the subsection “Identification of the TOPBP1-BRCT1,2 binding site in RHNO1”.

2) Please show the Mdb1 pS216, pT218 peptide sequence in Figure 5. Perhaps the sequence is obvious that it doesn't bind BRCT1,2 based on the prediction.

We have included the sequence in the figure legend, and have added a list of all fluorescent peptides used to the Materials and methods.

3) The symbol legends for all fluorescence polarisation curves are not displayed correctly. Please specify what ND stands for.

This was a PDF conversion problem that has been rectified – we have added a note in the Materials and methods to indicate that ND signifies a dose response for which a binding curve and Kd could Not be Determined.

4) In Figure 5C, please explain "inaccurate" and how 20-fold reduction (subsection “Identification of a novel Rad4-BRCT1,2 binding site in Mdb1”, fourth paragraph) was derived when comparing the Kd of BRCT1,2 WT and K151E mutant for Mdb1 pT113 peptide.

We used ND (inaccurate) to indicate that while the indicated dose response could be fitted to a single-site model curve, it doesn’t saturate and the quality of fit is so poor that any Kd derived is very inaccurate. This is rather confusing, and so we have removed the fitted curve and now show the data points only as Not Determined (ND). The 20-fold is a reference to the lower end of the range of Kds that can be extracted from this inaccurate data. As we can’t assign a reliable value to this weak dose response, we have amended the text in the fourth paragraph of the subsection “Identification of a novel Rad4-BRCT1,2 binding site in Mdb1”, to be qualitative rather than quantitative.